# Screening of Native *Trichoderma* Species for Nickel and Copper Bioremediation Potential Determined by FTIR and XRF

**DOI:** 10.3390/microorganisms11030815

**Published:** 2023-03-22

**Authors:** Gordana Racić, Igor Vukelić, Branko Kordić, Danka Radić, Milana Lazović, Ljiljana Nešić, Dejana Panković

**Affiliations:** 1Faculty of Ecological Agriculture, Educons University, Vojvode Putnika 87, 21208 Sremska Kamenica, Serbia; 2Faculty of Natural Sciences, University of Novi Sad, Trg Dositeja Obradovića 4, 21000 Novi Sad, Serbia; 3Institute of General and Physical Chemistry, Studentski Trg 12-16, 11158 Belgrade, Serbia; 4AbioTech Lab, Vojvode Putnika 87, 21208 Sremska Kamenica, Serbia; 5Faculty of Agriculture, University of Novi Sad, Trg Dositeja Obradovića 8, 21000 Novi Sad, Serbia

**Keywords:** biosorption, bioaccumulation, nondestructive methods, heavy metals, *Trichoderma*

## Abstract

Soil pollution with heavy metals is a serious threat to the environment. However, soils polluted with heavy metals are considered good sources of native metal-resistant *Trichoderma* strains. *Trichoderma* spp. are free-living fungi commonly isolated from different ecosystems, establishing endophytic associations with plants. They have important ecological and biotechnological roles due to their production of a wide range of secondary metabolites, thus regulating plant growth and development or inducing resistance to plant pathogens. In this work we used indigenous *Trichoderma* strains that were previously isolated from different soil types to determine their tolerance to increased copper and nickel concentrations as well as mechanisms of metal removal. The concentrations of bioavailable metal concentrations were determined after extraction with diethylene-triamine pentaacetate (DTPA)-extractable metals (Cd, Cr, Co, Cu, Pb, Mn, Ni, and Zn) from the soil samples by inductively coupled plasma-optical emission spectrometry (ICP-OES). Two indigenous *T. harzianum* strains were selected for copper tolerance, and three indigenous *T. longibrachiatum* strains were selected for nickel tolerance tests. Strains were isolated from the soils with the highest and among the lowest DTPA-extractable metal concentrations to determine whether the adaptation to different concentrations of metals affects the mechanisms of remediation. Mechanisms of metal removal were determined using Fourier-transform infrared spectroscopy (FTIR) and X-ray fluorescence spectroscopy (XRF), non-destructive methods characterized by high measurement speed with little or no need for sample preparation and very low costs. Increased DTPA-extractable metal content for nickel and copper was detected in the soil samples above the target value (TV), and for nickel above the soil remediation intervention values (SRIVs), for total metal concentrations which were previously determined. The SRIV is a threshold of metal concentrations indicating a serious soil contamination, thus confirming the need for soil remediation. The use of FTIR and XRF methods revealed that the presence of both biosorption and accumulation of metals in the *Trichoderma* cells, providing good bioremediation potential for Ni and Cu.

## 1. Introduction

Metals are natural constituents of ecosystems but if present in excess concentrations are a threat to the environment, humans, and other living organisms. Though of geological origin, in recent decades they have been released into soil, water, and air in large quantities through different industrial processes, agricultural activities, and bad waste management. According to the European Environmental Agency, contamination of sites with toxic and trace elements (Ag, As, Be, Cd, Cr, Cu, Hg, Ni, Pb, Sb, Se, Tl, and Zn) will rise by 50% by 2025 [1]. Some metals (copper, manganese, cobalt, zinc, and chromium) are considered essential and when present in low concentrations play important roles in metabolic activities of living organisms [2]. However, when nickel, chromium, copper, zinc, mercury, lead, and cadmium are present in excess, they are known as toxic to microorganisms, plants, and humans, also causing various adverse impacts on ecosystems. Some metals originating from different industrial processes are water-soluble, thus being easily transported to soil and groundwater systems, causing their magnification [3]. High concentrations found in agricultural soils due to application of different chemicals which are used for plant protection lead to deleterious effects on plants and ecosystem biodiversity, and, in cases of long-term exposure, human health is also affected [4].

Removal of heavy metals with traditional physicochemical remediation techniques is time-consuming and expensive [4]. In recent years, bioremediation techniques, which include the use of microorganisms or plants for cleaning up the environment, have been exploited, as they are cost-effective and environmentally friendly [5]. Due to their ability to survive under different environmental conditions and extensive branching of mycelia, fungal species have been intensively investigated in terms of degrading organic pollutants and heavy metals. Heavy metal accumulation in the environment is difficult to manage as metals cannot be metabolized and are not biodegradable. The role of microorganisms in surpassing heavy metal stress is based on various processes, such as precipitation, adenosine triphosphate (ATP)-mediated efflux, biosorption, alteration in cellular morphology, and change of ionic states of metals, thus affecting metal solubility, mobility, and bioavailability [6,7,8,9,10].

The use of bioremediation is dependent on the metabolic activity of selected microorganisms, redox state of metal, and environmental conditions [11]. The response of fungi to the pollution depends on the concentration of heavy metals, type of metal, the nature of the medium, and the used species [12]. Some of the species previously reported as capable of degradation of heavy metals and organic pollutants are *Rhizopus oryzae*, *Trichoderma* spp., *Aspergillus* spp., *Penicillium chrysogenum*, and *Gloeophyllum sepiarium* [13]. For example, accumulation of As by *Trichoderma asperellum* SM-12F1, *Penicillium janthinellum* SM-12F4, and *Fusarium oxysporum* CZ-8F1 was reported [14]. Indigenous microorganisms are considered to be the most effective for the in situ bioremediation, due to their potential to survive in extreme conditions of abiotic stress. Therefore, they are considered a better choice compared to non-indigenous microorganisms. Thus, ecosystems polluted with heavy metals are considered good sources of indigenous metal-resistant microorganisms [15]. Usually, one or more mechanisms are involved in metal tolerance of microbes. It is believed that biosorption is the most common mechanism [16,17,18,19], but bioaccumulation [20], biotransformation [21], and biomineralization [22] are also well-developed mechanisms in bacteria and fungi for metal removal. Uptake of metals can be active and metabolism-dependent (bioaccumulation) or passive and metabolism-independent (biosorption) [23]. Biosorption mechanisms are based on physicochemical interactions between metal ions and functional groups of biomolecules of the microorganism cell wall. The composition and structure of the cell wall vary across different microorganisms. The cell wall of fungi consists of glucans, chitin, glycoproteins, and melanin [24]. The presence of negatively charged functional groups (e.g., carbonyl, phosphoryl, and hydroxyl groups) in membranes depends on the type of fungus, and their presence in membranes provides an interaction with positive metal ions [25,26].

Filamentous fungi that belong to the genus *Trichoderma* are found in all climate zones in soils rich in organic matter, as well as sandy and tropical soils, and can establish an endophytic interaction with plants. Today there are more than 300 identified *Trichoderma* species, many of which have not yet been formally described [27]. Due to their rapid growth, ability to compete for nutrients, the possibility of utilizing a variety of substrates, adaptation to environmental changes, metabolite production, and resistance to many pollutants, they are widely studied as beneficial fungal species. *Trichoderma* strains can be easily isolated, characterized, and multiplied in in vitro conditions and maintained for a long period of time at −20 °C while maintaining viability and properties. In terms of bioremediation potential, the possibility of *Trichoderma* species tolerating and accumulating some toxic contaminants is reported [28]. Degradation of trinitrotoluene (TNT) has been observed by *T. viride* [29]. Moreover, the reduction of hexavalent chromium by *T. inhamatum* was recorded [30]. Nongmaithem et al. [31] reported tolerance of *Trichoderma* strains to increased concentrations of nickel and cadmium.

Fungi that belong to the *Trichoderma* genus can tolerate higher copper concentrations in soil, up to 1200 mg/L [32], provided by physiological and genetic adaptations [33]. Copper accumulation is mostly found in vineyard soils due to the extensive use of copper (Cu)-based fungicides. In the LUCAS Topsoil database, out of 342 vineyard soil samples, 14.6% had a Cu concentration above the proposed threshold (100 mg kg^−1^), whereas the mean value was 49.26 mg kg^−1^, with very high variability between examined countries [34]. In our previous work, we determined Cu concentrations in 27 different locations, and it ranged from 8.148 to 72.820 mg kg^−1^ [35]. However, the highest were determined in the soil samples taken from a vineyard under conventional farming conditions, probably originating from the application of fungicides. Copper exists in two oxidation states (Cu(II) and Cu(I)) and when present in excess leads to the formation of reactive oxygen species, causing damage at the plant cellular level [36].

Nickel is naturally present in the soil in the form of organic compounds, sulfates, nitrates, and halides. However, due to extensive use in many industrial processes, it causes soil pollution. After being accumulated by plants from contaminated soil, nickel enters the food chain, causing serious damage to the human body [37].

In this work, we used indigenous *Trichoderma* strains that were previously isolated from different soil types [35] to assess their tolerance to copper and nickel concentrations ranging from 30 to 240 mg/L. Firstly, the bioavailable, i.e., DTPA-extractable, metal content was analyzed in examined soil types. Based on given results *Trichoderma* strains that were isolated from the soils with the highest and among the lowest concentrations were selected in order to determine whether the adaptation to different concentrations of metals affects the mechanisms of remediation. In order to determine mechanisms of metal removal by fungi, Fourier-transform infrared spectroscopy (FTIR) and X-ray fluorescence spectroscopy (XRF) were performed. The use of these methods revealed that strains examined in this study are promising for bioremediation of nickel and copper ranging from 30 to 240 mg/L by both biosorption and bioaccumulation.

## 2. Materials and Methods

### 2.1. Description of Soil Samples

Soil samples used in this study were collected from 14 different locations in Serbia, considering that most common soil types of the region are represented, located in polluted as well as non-polluted areas. In total, 23 soil samples were collected: agricultural soils (5) and vineyard soils (4), conventional and organic farm soils, and forest soils (14). Samples were randomly chosen from the top horizon (A) of soils at two depths 0–30 and 30–60 cm. The only location where the sample was taken from the 0–20 cm depth was at Zmajevac, soil sample number 11, because dominant parental rock was reached at 20 cm. Physical, chemical, and microbiological characteristics and total metal content of the samples were previously described and reported [35].

### 2.2. Determination of DTPA-Extractable Metal Concentration in Soil Samples

The concentrations of bioavailable metal concentrations were determined after extraction with diethylene-triamine pentaacetate (DTPA). Soil samples were extracted with 0.005 M DTPA solution in 0.1 M TEA (triethanolamine) buffer and 0.01 M calcium chloride, pH 7.3, as described in the work of [38]. The pH value of the prepared solution was adjusted with 1 mol/L of HCl. Ten grams of soil sample was quantitatively transferred to Erlenmayer flask, and 20 mL of DTPA solution (ratio 1:2, *w*/*v*) was added. Flasks were shaken in a shaker (OS 2000 open air dual shaker, Jeio Tech, Taejon-Jikhalsi, Republic of Korea) for 2 h at 180 rpm, and afterwards samples were filtered through filter paper (Munktell & Filtrak, diameter 110 mm, 84 g/m^2^) into plastic containers.

The samples were analyzed according to the EPA 6010c method on the ICP-OES system Thermo iCAP 6500 Duo. The content of the examined heavy metals (Cr, Co, Ni, Cu, Zn, Cd, Pb, and Mn) was determined after signal intensity readings at the wavelengths for which a given metal is most sensitive to response: cadmium (Cd)—228,802 nm; cobalt (Co)—228,616 nm; chromium (Cr)—267,716 nm; copper (Cu)—327,396 nm; manganese (Mn)—257,610 nm; nickel (Ni)—231,604 nm; lead (Pb)—220,353 nm; zinc (Zn)—213,856 nm. Trace Metal Standards I (TMS I) were used as standards.

### 2.3. Selection of Nickel- and Copper-Tolerant Isolates and Metal Tolerance Test of Trichoderma Spp.

Soil samples were dissolved in sterile water, mixed, and then diluted by 10^−1^, 10^−2^, 10^−3^, 10^−4^, and 10^−5^ times. The dilutions were cultured on Rose Bengal Medium [39]. After 48 h, *Trichoderma* colonies were extracted and cultured on potato dextrose agar (PDA) medium. Selected strains were identified based on their ITS sequences, as described previously [38]. Pure cultures of the selected strains were maintained and stored in 20% glycerol solution at −20 °C. Indigenous *Trichoderma* strains isolated from the soils with the highest and among the lowest DTPA-extractable concentrations were selected for further investigation to determine whether the adaptation to different concentrations of metals affects the mechanisms of remediation. Two *T. harzianum* strains were selected for copper tolerance (SZMC 20969-soil sample 1 and SZMC 20660-soil sample 17), and three *T. longibrachiatum* strains were selected for nickel tolerance (SZMC 22664-soil sample 7, SZMC 22669-soil sample 11, and SZMC 22665-soil sample 12). Fungal isolates were preincubated on potato dextrose agar (PDA) medium at 25 °C for 5 days.

The metal tolerance test was performed by measuring the diameter of fungal colonies on plates containing Ni (II)- and Cu (VI)-amended PDA medium and comparing the diameter with colonies on control PDA plates. PDA medium was autoclaved before metal addition and afterwards amended with 30, 60, 120, and 240 mg/L of Ni and Cu, in the form of salts NiCl_2_ × 6H_2_O and CuSO_4_ × 5H_2_O. These concentrations were selected to test fungal growth on concentrations above and below the maximum allowed values according to the EU legislation. According to the Dutch Target and Intervention Values (2000), the TV is 36 mg/L for Cu and 35 mg/L for Ni, and the SRVI value (standard soil) is 190 mg/L for Cu and 210 mg/L for Ni [40].

Fungal isolates were preincubated on PDA medium at 25 °C for 5 days. Afterwards, mycelia discs of 5 mm diameter were cut and placed on the center of the control and Ni (II) and Cr (VI)-enriched plates. Three replicate plates were used per treatment. Fungi were incubated in the dark at 25 °C for 48, 96, and 144 h. Fungal colony diameter was measured, and the percentage of inhibition calculated according to Abbott’s formula [41]:PI = (C − T)/C × 100
where PI is the percentage inhibition, C is the colony diameter (mm) on the control PDA plate, and T is the colony diameter (mm) on the Ni (II)- and Cr(VI)-amended PDA plates.

### 2.4. Morphological Characterization

Morphological characterization of *Trichoderma* strains grown at two concentrations (60 and 120 mg/L) of Cu and Ni was performed [42,43]. The length and width of the phialides and conidia were analyzed microscopically (Olympus microscope CX4, Tokyo, Japan; Bresser digital camera, 5.0 mp, Rhede, Germany). *Trichoderma* colonies were grown on PDA medium with or without Cu and Ni for 5 days, covered with water, and placed on microscope slides.

### 2.5. Fourier-Transform Infrared Spectroscopy (FTIR) and X-ray Fluorescence Spectroscopy (XRF) Analysis of Fungal Cells

Six days after the growth on either pure PDA medium or PDA medium amended with 60 mg/L of copper or nickel, fungal colonies were removed from the Petri dishes with an inoculation loop and placed (into a new petri dish) on the holder for immediate fast recordings on fresh mycelia by FTIR and XRF.

To identify the functional groups present in the biomass of *Trichoderma* before and after treatment with metal ions of copper and nickel, FTIR analysis was performed [44] using Alpha FTIR (Bruker Optics, Ettlingen, Germany) in attenuated total reflectance (ATR) measurement mode, with a diamond crystal in the mid-IR range 400/4000 cm^−1^ and resolution of 4 cm^−1^. For each FTIR spectrum a total of 24 scans were collected. Spectra were analyzed with OPUS software (Bruker, Germany).

In contrast, accumulation, and quantitative determination of examined metals inside the cells was determined with the XRF technique [45,46]. Recordings were carried out with X-ray fluorescence spectroscopy (ARTAX 200 µ-XRF, BRUKER Nano, Berlin, Germany) using the Rh source at 25 kV and 1.5 mA with an exposure time of 100 s. Spectra were analyzed with integrated software BRUKER ARTAX SPECTRA 7.

### 2.6. Statistical Analysis

Metal tolerance test and morphological characterization data were analyzed by linear regression (ANOVA) and Tukey’s post hoc test (*p*-value < 0.05) using GraphPad Prism software version 9.0.0 (GraphPad Software, San Diego, CA, USA).

## 3. Results and Discussion

Though mostly known as antagonists to different plant pathogens, species of *Trichoderma* have beneficial ecological and biological roles, which are widely used in various fields of biotechnology and agriculture. Some of the species are known to adapt and grow under various environmental conditions and are therefore present in different ecosystems. Their ability to grow in the presence of elevated metal concentrations was demonstrated previously [47]. Several authors have suggested the potential use of these fungi in the bioremediation of substrates contaminated with metals [48,49,50].

In our previous work we observed high concentrations of total Cr and Ni in the soil sample from Zmajevac. According to our previous results for these two metals (Cr 718 mg kg^−1^ and Ni 1587 mg kg^−1^), their concentration in the soil sample from Zmajevac significantly exceeds the SRIV [40]. In the same investigation, the highest concentrations of copper were determined in soil samples from vineyards. In this work, we determined DTPA-extractable, i.e., available, metal content from the soil of the same locations (Table 1).

Sample 11 (forest soil) contained maximum determined concentrations of Co, Mn, and Ni. The highest concentration of Cu was determined in soil sample 1 (agricultural soil), of Cd in sample 18 (agricultural soil), of Cr in sample 22 (agricultural soil), of Pb in sample 12 (forest soil), and of Zn in sample 16 (agricultural soil). The lowest concentrations of Pb, Ni, and Zn were determined in sample 10 (forest soil) of Cd in sample 12 (forest soil), of Cr in sample 4 (agricultural soil), of Cu in sample 13 (forest soil) and of Mn in sample 7 (agricultural soil). Increased DTPA-extractable metal content for nickel and copper was detected in the soil samples with above-remediation values for total metal concentrations, which was not the case for chromium.

Increased DTPA-extractable metal content for nickel and copper was detected in the soil samples above the TV, and for nickel above the SRIV, for total metal concentrations which were previously determined. The TV indicates the level at which there is a sustainable soil quality [40]. The SRIV is a threshold of metal concentrations indicating a serious soil contamination, thus confirming the need for soil remediation.

Two *T. harzianum* strains were selected to be tested for tolerance to copper (SZMC 20969 isolated from soil sample 1 and SZMC 20660 isolated from soil sample 17), and three *T. longibrachiatum* strains were selected for nickel tolerance (SZMC 22665, SZMC 22664, and SZMC 22669, isolated from soil samples 11, 7, and 12, respectively) (Table 1).

We focused on a selection of strains from soils with increased concentrations of copper and nickel; however, other metals were also measured to avoid synergistic effects of increased soil concentration of multiple heavy metals.

The copper concentration of 30 mg/L did not have a significant effect on *Trichoderma* strains. The concentration of 60 mg/L resulted in 24.9% growth inhibition for *T. harzianum* SZMC 20660. The inhibition was observed in the first 48 h; however, at later time points it was not inhibitory. After 48 h, the Cu concentration of 120 mg/L inhibited growth of *T. harzianum* SZMC 20660 and *T. harzianum* SZMC 20969 by 84% and 69%, respectively. At the same Cu concentration after 96 h, the growth inhibition was lower: *T. harzianum* SZMC 20660 growth was decreased by 28.6%, while *T. harzianum* SZMC 20969 growth was inhibited by 16%. However, the highest investigated concentration of 240 mg/L was inhibitory in all tested time intervals and ranged from 79% to 100% (Table 2).

The concentration of 30 mg/L Ni did not have significant effects on growth of *Trichoderma* strains. The inhibition effects of 60 mg/L were significant only in the first 48 h, while the concentration of 120 mg/L was inhibitory after 48 and 96 h. The concentration of 240 mg/L was inhibitory for all examined *Trichoderma* species at all time points (Table 3).

Although at the limit of statistical significance, the trend of growth stimulation with the lowest applied concentrations of metals was observed in our experiments. Stimulatory effects caused by low levels of potentially toxic agents on growth have often been observed in a range of taxa after exposure to a variety of agents [51]. It was shown that relatively low metal concentrations caused *T. virens* mycelia aggregations, enabling avoidance of metal toxic effects [52].

The statistical data were analyzed with the use of one-way ANOVA and Tukey’s post hoc test and a statistically significant difference (*p*-value < 0.05) is denoted by a different letter in the table.

All examined strains exhibited similar tolerance to increased metal concentrations, both from heavily polluted and unpolluted soils (Table 2 and Table 3). Tripathi [53] demonstrated high tolerance of selected *Trichoderma* isolates to increased concentrations of nickel, arsenate, and copper in the range of 100 to 250 mg/L. According to our results, *T. longibrachiatum* could tolerate a Ni concentration of 60 mg/L. The soil remediation value for nickel is 210 mg/L; however, the target maximal value is 35 mg/L. The examined concentration of 60 mg/L exceeds the limits of the target value by almost two times; therefore, we considered the examined strain as a promising bioremediation agent for this metal. In the case of increased copper, strain *T. harzianum* 20,969 was able to grow at 120 mg/L of Cu. The soil remediation value for copper is 190 mg/L, but the target maximal value is 36 mg/L, thus making this strain a promising candidate for bioremediation process of this metal from the soils.

### 3.1. Micro Morphological Characteristics

In order to determine the effect of metals on micro-morphological characteristics of fungi, the size of phialides and conidia was measured. The conidia size of *T. harzianum* ranged from 2.4–3.2 × 2.2–2.8 μm, while *T. longibrachiatum* ranged from 3.6–6.5 × 2.2–3 μm [42]. The size of the lateral phialides of *T. harzianum* ranged from 5–7 × 3–3.5 μm, while the dimensions of the apical phialides varied around 18 × 2.5 μm. The size of the *T. longibrachiatum* phialides ranged from 6–14 × 2.5–3 μm.

Based on more than 400 measurements of morphological characteristics of examined *Trichoderma* spp. strains, which included the dimensions of conidia and phialides, the influence of different concentrations of heavy metals Cu and Ni on their morphology was not observed (Appendix A). Good growth of fungi in combination with normal phialides and conidia morphologies is considered an indicator of highly tolerant strains to increased heavy metal concentrations [53].

Küpper [54] observed different copper levels in examined *Trichoderma* strains. Among them, *T. koningiopsis* was a highly tolerant isolate showing no deformations of mycelia based on microscopical investigations, similar to our results described in Appendix A.

### 3.2. FTIR Analysis

To investigate the effects that Ni and Cu ions can have on the fungal cells, FTIR analysis of the untreated cells and the cells exposed to the metal solutions was conducted. FTIR spectroscopy has been extensively used for the investigation of fungi biochemical structure [55,56,57,58].

The broad signal at 3287.59 cm^−1^ corresponds to stretching vibrations of O-H and N-H bonds from different macromolecules (Figure 1). Bands at 2924 and 2854 cm^−1^ originate from C-H symmetric and asymmetric stretching vibrations. The band at 1744 cm^−1^ can be assigned to the C=O stretching vibration from ester groups in lipids. The amide I band originating from C=O and C-N stretching vibrations from proteins appears at 1639 cm^−1^, while the amide II band corresponding to N-H and C-N bending vibrations can be found at 1545 cm^−1^. The complex group of bands at 1456, 1415, 1378, and 1314 cm^−1^ can be attributed to skeletal vibrations and bending modes of O-H, CH_2_, and CH_3_ groups originating from proteins and lipids. The band at 1235 cm^−1^ can be assigned to the overlapping of the bands corresponding to PO_2_ and C-O-C asymmetric stretching vibrations, originating from phosphate compounds and polysaccharides. Further, the band at 1153 cm^−1^ was assigned to C-O-C symmetrical stretching vibration from polysaccharides. The band at 1077 cm^−1^ with a shoulder forming on the higher wavenumbers side of the band probably originates from overlapping of the bands of the PO_2_ symmetric vibrations from phosphate compounds and the band of C-O stretching coupled with C-O bending from cell carbohydrates. The strong band at 1029 cm^−1^ was assigned to C-O stretching, asymmetrically coupled with adjacent C-C stretching from the ester groups in the glycerides.

A comparison of the band positions between *Trichoderma* control cells and the *Trichoderma* strains grown in the presence of metals are presented in Table 4. Stacked FTIR spectra of control cells and cells of *Trichoderma* strains grown in the presence of Ni are given in Figure 2. From the spectra, it can be observed that the presence of Ni influenced the change in position, shape, and relative intensities of bands in the spectra of fungi strains compared to the control sample. The most distinguishable differences are in spectral regions of around 2930–2850, 1744, 1460–1310, and ~1030 cm^−1^ which are assigned to lipid functional groups.

In the region of 2930–2850 cm^−1^, which corresponds to the C-H vibration, changes in the relative intensities of bands and their position, compared to the control sample spectrum, were observed, particularly in the spectrum of *T. longibrachiatum* 22665. Further, the relative intensity of the C=O stretching vibration decreased in the samples containing Ni ions. Changes in shape and position of the peaks of the bending vibrations of O-H, CH_2_, and CH_3_ groups in the region of around 1460–1310 cm^−1^ were observed. The above observations suggest that the metal ions influence a fungal cell wall through interaction with lipid molecules. A change in the shape of the band at 1077 cm^−1^ and the amide II band can be observed, which corresponds to functional groups in proteins and phosphate compounds. The change in these bands implies that the Ni ions also interact with the glycoproteins and phosphoryl compounds. This agrees with the literature, as carboxyl and phosphoryl groups have been suggested as particularly important for the reactivity of cell walls toward metals [59]. A decrease in the relative intensity of the band at 1744 wavenumbers compared to the control sample was observed for all strains in the presence of nickel. In strain 22665, which was the most susceptible to the influence of metal ions, this band appears as a barely distinguishable shoulder of the band at 1637 wavenumbers. The band at 1744 wavenumbers was assigned to C=O stretching vibration from ester groups in lipids. A possible explanation for the influence of nickel on the band’s intensity at 1744 wavenumbers is the formation of coordination bonds of metal ions with the ester groups and the overall structure of lipids.

Stacked FTIR spectra of the control cells and cells of *Trichoderma* strains grown in the presence of Cu are given in Figure 3. Biosorption of Cu ions is known to have an impact on cell wall morphology and can cause severe damage to the cell wall [60]. A comparison of the FTIR spectra of the control sample and the *Trichoderma* strains grown in the presence of Cu shows similar changes to those previously observed in the samples with Ni ions. Changes were noticed in the regions of the CH_2_ and CH_3_ stretching and bending vibrations, at approximately 2930–2850 and 1460–1310 cm^−1^, respectively. The relative intensity of the C=O stretching vibration at 1744 cm^−1^ decreased in the samples containing Cu ions. Moreover, the relative intensity of the peak at ~1030 cm^−1^ assigned to the C-O stretching vibration coupled to adjacent C-C vibration decreased in the spectra of samples with Cu ions. A change in the relative intensity was also observed for the amide II band and the band at 1077 cm^−1^. These observations suggest that the Cu interacts with the cell wall, probably through interactions with polar groups of lipide, protein, and polysaccharide macromolecules.

Kumar and Dwived [61] obtained a shift in IR spectra of *T. lixii* CR700 biomass grown in 100 mg/L of Cu^2+^ PDB medium indicating the role of the COO─ functional group in surface biosorption and accumulation of Cu^2+^ inside the cell of CR700 after treatment with Cu through electrostatic attraction and the involvement of protein and other protein derivatives. Tu [62] used copper-tolerant strain *Gibberella* sp. NT-1 to examine mechanisms for Cu^2+^ removal and revealed that the binding of Cu (II) to NT-1 was mainly achieved by surface adsorption through forming polydentate complexes with carboxylate and amide groups and forming Cu–nitrogen-containing heterocyclic complexes through Cu(II)–π interaction. FTIR spectra of strain *T. brevicompactum* QYCD-6 grown in the presence of 50 mg/L of different metals (Cu(II), Cr(VI), Cd(II), Pb(II), and Zn(II) indicated the involvement of diverse functional groups, including amino, hydroxyl, carbonyl, phosphoryl, and nitro groups, thus suggesting this strain as a promising candidate for the remediation of heavy-metal polluted sites [25]. Sujatha et al. [63] investigated FTIR spectra of free and nickel-sorbed biomass of *T. viride*, suggesting that amine, hydroxyl, C=O, and alcoholic C–O groups of biomass could be involved in the biosorption of nickel ion onto *T. viride*.

The accumulation of nickel and copper in fungal cells was observed by XRF analysis (Table 5), which allows the determination of the relative concentration of heavy metals within the cells, with a spatial resolution of 0.2 µm and sensitivity below ppm [64].

As expected, neither nickel nor copper were found in control samples. However, in samples where nickel was applied in the medium, it was found that the content of nickel in the cells of *T. longibrachiatum* SZMC 22665, *T. longibrachiatum* SZMC 22669, and *T. longibrachiatum* SZMC 22664 was 0.249%, 0.171%, and 0.103% respectively. The content of copper in cells of *T. harzianum* SZMC 20969 was higher (0.070%) in comparison to *T. harzianum* SZMC 20660 (0.021%).

Küpper [54] observed different copper levels in examined *Trichoderma* strains. Different accumulation of our strains might be explained by copper transport mechanisms previously described in yeast [65]. Copper chaperones and ATPases are part of the metal secretory pathway. After being trapped, metal ions are transported into the extracellular environment via the plasma membrane [54,63].

In general, fungi respond in various ways when being exposed to raised levels of metal concentrations. Mostly, metals are either accumulated inside the fungal cell or are adsorbed on the fungal cell surface due to the presence of functional groups [61]. Accordingly, one of the mechanisms that fungi use to cope with heavy metal toxicity is their binding to functional groups of proteins, polysaccharides, lipids, and other organic ligands of cell walls, thus reducing element ion toxicity exerted on cells. In addition, glucans, polymers, and chitin (containing chitosan), as components of the *Trichoderma* spp. cell walls, affect high metal biosorption by this species [54]. The reduction of copper by the cell-wall surface metalloreductases of *Saccharomyces*, which was later transported using high-affinity membrane-associated transporters through the plasma membrane, was also reported [65]. Küpper [54] suggests that a similar pattern can be adopted for copper-tolerant *Trichoderma* strains. Anand [66] investigated biosorption of copper by *Trichoderma* strains in liquid culture and reported 17% of 100 mg/L of Cu removal from the liquid media at 30 °C and 72 h of incubation. Yazdani [67] reported 10.93% of Cu removal by *T. atroviride* at seven days of incubation. The same authors suggested better biosorption capacity of strains isolated from the substrates where the elevated concentrations of heavy metals were determined. One of the important factors is the incubation period, which affects the growth and metabolic activities of the applied organism [68]. As the incubation period increases, the biomass of the fungus and thus the rate of removal of heavy metals from appropriate media also increase [69]. 

Moreover, cell-wall sorption of metals, such as Cr (VI) on *T. lixii* CR700 [70] and Cu (II) biosorption by NT-1 strain [70], have been reported as a mechanism in removal of these pollutants [67].

Our results indicate the presence of both biosorption and accumulation of metals in the *Trichoderma* cells, providing them good bioremediation potential for Ni and Cu.

Biotechnological strategies which exploit the tolerance and properties of plants and microorganisms (bacteria, microalgae, yeast, and fungi) for the detoxification and stabilization of heavy metals have emerged as new and innovative technologies and show increasing possibilities for the restoration of contaminated soils. Bioremediation and phytoremediation are reliable, cost-effective, efficient, and environmentally feasible alternatives. However, phytoremediation alone can be slow.

It was shown recently that bioremediation could be combined with phytoremediation for effective recultivation [71]. Characterization and selection of genera associated with the plant microbiome with the potential for bacteria-assisted phytoremediation of soils contaminated with heavy metals was proposed as the most efficient [72].

It would be worthwhile to examine the interaction of *Trichoderma* strains examined in this study with metal accumulating plants, such as sunflower [73]. *Trichoderma* strains with the ability to promote metal uptake by host plants would provide irreversible removal of metals from the soil.

## 4. Conclusions

Examined *Trichoderma* species could grow in the presence of copper and nickel, in the range from 30 to 240 mg/L. FTIR spectra of cells grown in Ni presence imply that the Ni ions interact with the lipid molecules, glycoproteins, and phosphoryl compounds. However, stacked FTIR spectra of the control cells and cells of *Trichoderma* strains grown in the presence of Cu suggest that the Cu interacts with the cell wall, probably through interactions with polar groups of lipide, protein, and polysaccharide macromolecules. The XRF method revealed the accumulation of both Cu and Ni in the examined *Trichoderma* strains. The presence of both biosorption and accumulation mechanisms for metals removal by *Trichoderma* cells makes them promising candidates for bioremediation of soils contaminated with copper and nickel.

## Figures and Tables

**Figure 1 microorganisms-11-00815-f001:**
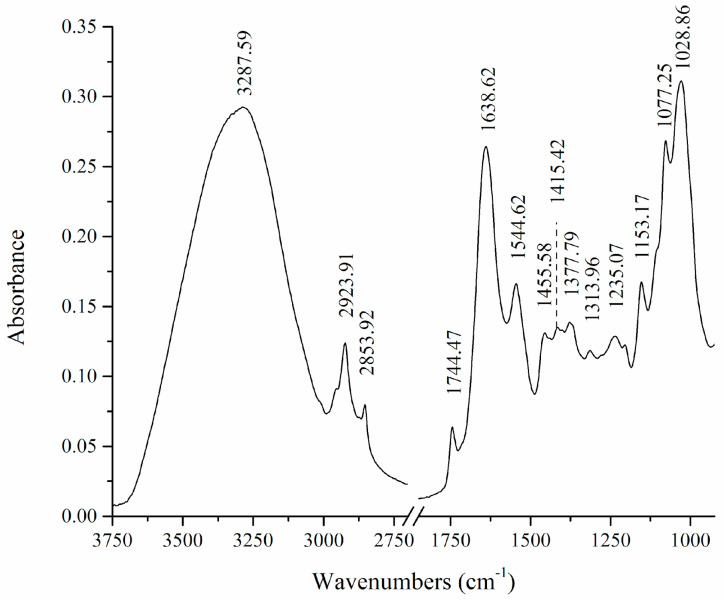
FTIR spectrum of untreated *Trichoderma* cell.

**Figure 2 microorganisms-11-00815-f002:**
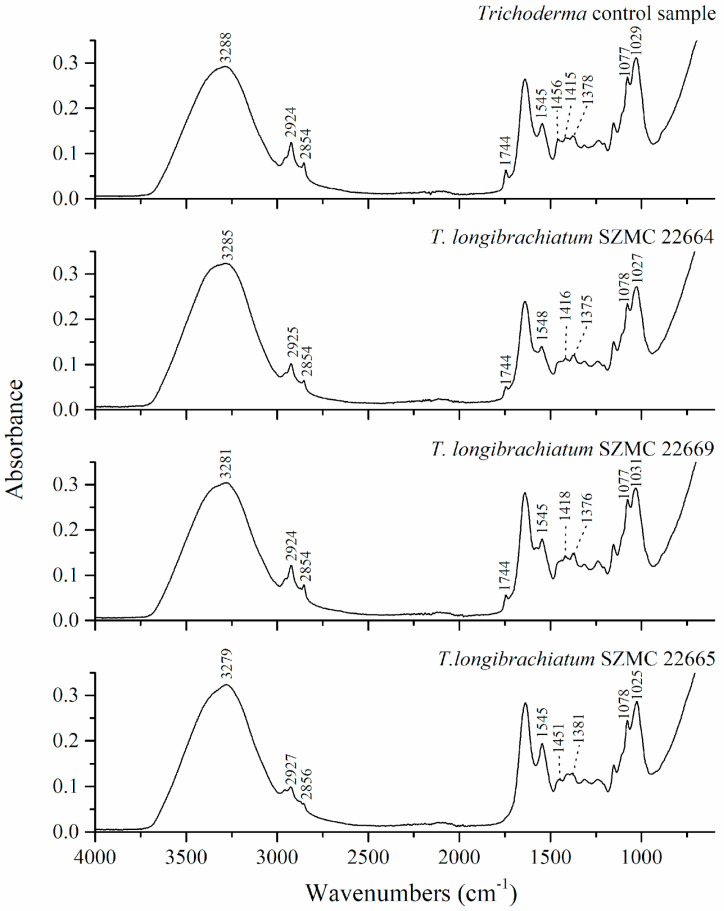
Stacked FTIR spectra of *Trichoderma* control cell group and strains 22664, 22669, and 22665 grown in 60 mg/L PDA medium of Ni.

**Figure 3 microorganisms-11-00815-f003:**
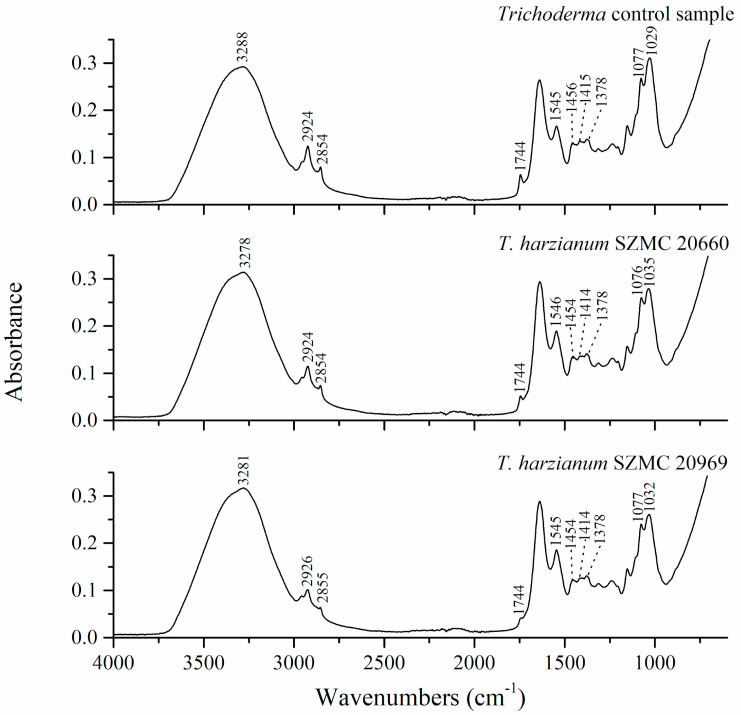
Stacked FTIR spectra of *Trichoderma* control cell group and strains 20660 and 20969 grown in 60 mg/L PDA medium of Cu.

**Table 1 microorganisms-11-00815-t001:** DTPA-extractable metal content in soil samples from different locations as described in the footer of the table (maximum concentrations are underlined and minimum in bold font). Results are presented as concentration in mg/kg ± STDEV.

Sample Number		Cd (mg/kg)	Cr (mg/kg)	Co (mg/kg)	Cu (mg/kg)	Pb (mg/kg)	Mn (mg/kg)	Ni (mg/kg)	Zn (mg/kg)
**1**	**Vineyard**	0.162 ± 0.012	0.019 ± 0.000	0.066 ± 0.002	19.684 ± 1.024	1.985 ± 0.083	18.268 ± 0.384	0.391 ± 0.038	5.026 ± 0.226
**2**	**Vineyard**	0.141 ± 0.007	0.019 ± 0.001	0.083 ± 0.003	16.718 ± 0.953	1.965 ± 0.100	20.122 ± 0.563	0.413 ± 0.032	4.348 ± 0.209
**3**	**Vineyard**	0.097 ± 0.008	0.018 ± 0.001	0.094 ± 0.002	11.818 ± 0.508	1.554 ± 0.071	22.970 ± 0.528	0.311 ± 0.026	2.294 ± 0.115
**4**	**Vineyard**	0.099 ± 0.005	0.017 ± 0.001	0.082 ± 0.002	12.466 ± 0.362	1.658 ± 0.066	21.914 ± 0.636	0.281 ± 0.019	2.166 ± 0.109
**5**	**Agricultural soil**	0.067 ± 0.002	0.018 ± 0.000	0.093 ± 0.003	1.507 ± 0.077	1.142 ± 0.061	30.554 ± 0.552	0.944 ± 0.072	1.023 ± 0.042
**6**	**Agricultural soil**	0.062 ± 0.003	0.018 ± 0.000	0.072 ± 0.002	1.087 ± 0.081	1.023 ± 0.052	26.736 ± 0.668	0.871 ± 0.081	0.994 ± 0.039
**7**	**Agricultural soil**	0.102 ± 0.004	0.142 ± 0.004	0.049 ± 0.001	3.552 ± 0.238	3.536 ± 0.163	8.842 ± 0.141	1.613 ± 0.147	1.268 ± 0.062
**8**	**Agricultural soil**	0.116 ± 0.005	0.095 ± 0.003	0.047 ± 0.002	3.550 ± 0.090	3.066 ± 0.123	9.428 ± 0.226	1.981 ± 0.184	1.315 ± 0.075
**9**	**Forest soil**	0.052 ± 0.002	0.024 ± 0.001	0.044 ± 0.002	1.837 ± 0.063	0.816 ± 0.026	10.228 ± 0.277	0.290 ± 0.016	0.667 ± 0.031
**10**	**Forest soil**	0.035 ± 0.003	0.026 ± 0.000	0.052 ± 0.001	1.472 ± 0.059	0.661 ± 0.036	9.158 ± 0.148	0.232 ± 0.014	0.471 ± 0.029
**11**	**Forest soil**	0.129 ± 0.006	0.237 ± 0.005	**2.130 ± 0.062**	1.114 ± 0.033	2.328 ± 0.109	**54.606 ± 0.983**	**106.056 ± 8.062**	2.722 ± 0.128
**12**	**Forest soil**	0.011 ± 0.001	0.491 ± 0.013	1.230 ± 0.038	0.625 ± 0.033	**8.257 ± 0.330**	24.912 ± 0.473	6.747 ± 0.614	3.001 ± 0.159
**13**	**Forest soil**	0.048 ± 0.002	0.305 ± 0.007	1.173 ± 0.028	0.556 ± 0.029	6.657 ± 0.280	13.488 ± 0.351	5.883 ± 0.565	2.383 ± 0.101
**14**	**Agricultural soil**	0.069 ± 0.005	0.041 ± 0.001	0.084 ± 0.004	1.696 ± 0.086	1.982 ± 0.097	21.766 ± 0.588	1.182 ± 0.050	0.637 ± 0.020
**15**	**Forest soil**	0.067 ± 0.003	0.061 ± 0.002	0.078 ± 0.002	2.066 ± 0.077	2.678 ± 0.088	24.022 ± 1.081	1.541 ± 0.149	0.621 ± 0.051
**16**	**Forest soil**	0.046 ± 0.004	0.018 ± 0.001	0.077 ± 0.003	7.326 ± 0.300	1.196 ± 0.049	24.020 ± 0.576	0.684 ± 0.065	**5.584 ± 0.123**
**17**	**Forest soil**	0.047 ± 0.003	0.018 ± 0.001	0.066 ± 0.003	5.478 ± 0.186	1.110 ± 0.057	20.438 ± 0.593	0.679 ± 0.054	4.390 ± 0.320
**18**	**Forest soil**	**0.209 ± 0.015**	0.026 ± 0.001	0.074 ± 0.003	1.560 ± 0.083	1.868 ± 0.077	26.102 ± 0.548	1.080 ± 0.077	0.729 ± 0.041
**19**	**Forest soil**	0.065 ± 0.004	0.359 ± 0.009	0.076 ± 0.002	1.595 ± 0.062	1.746 ± 0.079	28.582 ± 1.003	1.112 ± 0.082	0.774 ± 0.038
**20**	**Forest soil**	0.071 ± 0.005	0.366 ± 0.011	0.037 ± 0.001	1.023 ± 0.053	1.359 ± 0.053	13.722 ± 0.563	0.942 ± 0.071	1.558 ± 0.064
**21**	**Forest soil**	0.054 ± 0.004	0.018 ± 0.000	0.070 ± 0.002	0.962 ± 0.036	1.631 ± 0.078	21.184 ± 0.657	0.769 ± 0.066	1.188 ± 0.056
**22**	**Forest soil**	0.044 ± 0.002	**0.800 ± 0.033**	0.206 ± 0.005	3.406 ± 0.143	2.260 ± 0.081	36.716 ± 1.248	4.592 ± 0.445	0.811 ± 0.048
**23**	**Forest soil**	0.061 ± 0.004	0.078 ± 0.002	0.032 ± 0.001	4.328 ± 0.234	2.216 ± 0.112	12.824 ± 0.321	1.358 ± 0.126	5.518 ± 0.386
**MIN**		0.011 ± 0.001	0.017 ± 0.001	0.032 ± 0.001	0.556 ± 0.029	0.661 ± 0.036	8.842 ± 0.141	0.232 ± 0.014	0.471 ± 0.029
**MAX**		**0.209 ± 0.015**	**0.800 ± 0.033**	**2.130 ± 0.062**	**19.684 ± 1.024**	**8.257 ± 0.330**	**54.606 ± 0.983**	**106.056 ± 8.062**	**5.584 ± 0.226**

Sremski Karlovci 1 (**1.** 0–30 and **2.** 30–60 cm); Sremski Karlovci 2 (**3.** 0–30 and **4.** 30–60 cm); Titelski breg (**5.** 0–30 and **6.** 30–60 cm); Lok (**7.** 0–30 and **8.** 30–60 cm); Kaćka šuma (**9.** 0–30 and **10.** 30–60 cm); Zmajevac (**11.** 0–20 cm); Vrdnik (**12.** 0–30 and **13.** 30–60 cm); Kisač (**14.** 0–30 and **15.** 30–60 cm); Ljutovo (**16.** 0–30 and **17.** 30–60 cm); Rimski Šančevi (**18.** 0–30 and **19.** 30–60 cm); Čenej (**20.** 0–30 cm); Crepaja (**21.** 0–30 cm); Svilajnac 1 (**22.** 0–30 cm); Svilajnac 2 (**23.** 0–30 cm).

**Table 2 microorganisms-11-00815-t002:** Colony diameter of *T. harzianum* SZMC20660 and SZMC20969 grown on different copper concentrations. Different letters (a, b, c, d) indicate statistically significant differences according to Tukey’s test.

Copper mg/L	Time	SZMC20660 *T. harzianum*	SZMC20969 *T. harzianum*
Colony diameter (cm)			
Control	48 h	7.78 ± 0.19 ^a^	6.64 ± 0.99 ^ab^
30 mg/L		6.94 ± 0.08 ^a^	7.54 ± 0.18 ^a^
60 mg/L		5.88 ± 0.46 ^b^	6.04 ± 0.28 ^b^
120 mg/L		1.24 ± 0.55 ^c^	2.06 ± 0.08 ^c^
240 mg/L		0.00 ± 0.00 ^d^	0.00 ± 0.00 ^d^
Control	96 h	8.46 ± 0.05 ^a^	8.48 ± 0.05 ^a^
30 mg/L		8.46 ± 0.08 ^a^	8.44 ± 0.08 ^a^
60 mg/L		8.45 ± 0.07 ^a^	8.47 ± 0.04 ^a^
120 mg/L		6.04 ± 0.04 ^c^	7.12 ± 0.50 ^b^
240 mg/L		1.02 ± 0.02 ^d^	1.01 ± 0.05 ^d^
Control	144 h	8.50 ± 0.07 ^a^	8.52 ± 0.04 ^a^
30 mg/L		8.50 ± 0.07 ^a^	8.48 ± 0.05 ^a^
60 mg/L		8.49 ± 0.05 ^a^	8.50 ± 0.08 ^a^
120 mg/L		8.46 ± 0.05 ^a^	8.51 ± 0.05 ^a^
240 mg/L		1.78 ± 0.17 ^b^	1.50 ± 0.06 ^c^

**Table 3 microorganisms-11-00815-t003:** Colony diameter of *T. longibrachiatum* SZMC 20664, SZMC 20665, and SZMC 20669 grown on different nickel concentrations. Different letters (a, b, c, d, e) indicate statistically significant differences according to Tukey’s test.

Nickel mg/L	Time	SZMC20664 *T. longibrachiatum*	SZMC20669 *T. longibrachiatum*	SZMC20665 *T. longibrachiatum*
Colony diameter (cm)				
Control	48 h	7.60 ± 0.25 ^ab^	7.52 ± 0.19 ^ab^	7.50 ± 0.35 ^ab^
30 mg/L		7.70 ± 0.2 ^ab^	7.74 ± 0.25 ^ab^	7.83 ± 0.29 ^a^
60 mg/L		6.80 ± 0.20 ^c^	7.12 ± 0.20 ^bc^	6.88 ± 0.2 ^c^
120 mg/L		1.82 ± 0.31 ^d^	1.62 ± 0.37 ^d^	1.86 ± 0.71 ^d^
240 mg/L		0.06 ± 0.13 ^e^	0.00 ± 0.00 ^e^	0.00 ± 0.00 ^e^
Control	96 h	7.96 ± 0.20 ^a^	8.02 ± 0.22 ^a^	8.06 ± 0.23 ^a^
30 mg/L		7.92 ± 0.21 ^a^	7.98 ± 0.17 ^a^	8.04 ± 0.16 ^a^
60 mg/L		7.94 ± 0.19 ^a^	8.01 ± 0.18 ^a^	8.04 ± 0.11 ^a^
120 mg/L		5.44 ± 0.32 ^b^	5.46 ± 0.39 ^b^	5.94 ± 0.48 ^b^
240 mg/L		0.00 ± 0.00 ^c^	0.00 ± 0.00 ^c^	0.00 ± 0.00 ^c^
Control	144 h	8.12 ± 0.31 ^ab^	8.28 ± 0.24 ^a^	8.32 ± 0.13 ^a^
30 mg/L		8.08 ± 0.22 ^ab^	8.24 ± 0.16 ^a^	8.28 ± 0.22 ^a^
60 mg/L		8.07 ± 0.31 ^ab^	8.26 ± 0.21 ^a^	8.30 ± 0.12 ^a^
120 mg/L		7.68 ± 0.25 ^b^	8.00 ± 0.25 ^ab^	7.10 ± 0.20 ^c^
240 mg/L		0.00 ± 0.00 ^d^	0.00 ± 0.00 ^d^	0.00 ± 0.00 ^d^

**Table 4 microorganisms-11-00815-t004:** Wavenumbers (cm^−1^) and assignments of FTIR bands for *Trichoderma* fungal cells grown on clear PDA (control) and the PDA containing 60 mg/L Ni or Cu solutions.

	Ni			Cu			
Control (Untreated Cells)	*T. long.* 22664	*T. long.* 22669	*T. long.* 22665	*T. harz.* 20660	*T. harz.* 20969	Functional Group	Macromolecule
3288	3285	3281	3279	3278	3281	O-H, N-H	various
2924, 2854	2925, 2854	2924, 2854	2927, 2856	2924, 2854	2926, 2855	C-H	lipids
1744	1744	1744	/	1744	1744	C=O	lipids
1639	1639	1637	1637	1638	1638	Amide I C=O, C-N	proteins
1545	1548	1545	1545	1546	1545	Amide II N-H, C-N	proteins
1456, 1415, 1378, 1314	/, 1416, 1375, 1314	/, 1418, 1376, 1314	1451, /, 1381, 1313	1454, 1414, 1378, 1313	1454, 1414, 1378, 1313	O-H, CH_2_, CH_3_	lipids, proteins
1235	1240	1239	1241	1237	1240	PO_2_, C-O-C	phosphate compounds, polysaccharides
1153	1152	1154	1152	1154	1153	C-O-C	polysaccharides
1077	1078	1077	1078	1076	1077	PO_2_, C-O	phosphate compounds, polysaccharides
1029	1027	1031	1025	1035	1032	C-O, C-C	lipids

**Table 5 microorganisms-11-00815-t005:** XRF analysis for *Trichoderma* fungal cells grown on clear PDA (control) and the PDA containing 60 mg/L Ni or Cu solutions.

	Sample	Ni (%)	Cu (%)
	Control	/	/
	*T. longibrachiatum* SZMC 22665	0.249	/
60 mg/L Ni	*T. longibrachiatum* SZMC 22669	0.171	/
	*T. longibrachiatum* SZMC22664	0.103	/
	*T. harzianum* SZMC 20660	/	0.021
60 mg/L Cu	*T. harzianum* SZMC 20969	/	0.070

## Data Availability

Not applicable.

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
