# Peer review of "Screening of Native Trichoderma Species for Nickel and Copper Bioremediation Potential Determined by FTIR and XRF"

_microorganisms, 2023, doi:10.3390/microorganisms11030815_

Round 1

Reviewer 1 Report

GENERAL COMMENT

The study was made in the very relevant field of remediation of soil contaminated with heavy metals. The search for microbial strains capable of removing various pollutants from soil and water is a very popular theme. In this aspect, the study is important and relevant, and its results may provide some practical application of the studied strain in the future. However, the description of the methods used in the study and the presentation of results should be improved prior the manuscript acceptance (see comments below).

INTRODUCTION

Lines 87-97: actually, this text is not needed in the Intro section. Method description could be placed in the Materials and Methods section, while the last sentence describes the obtained results.

MATERIALS AND METHODS

 Line 100. It would be good to give a brief information about the choice of sampling sites. Were they chosen using any data on their contamination with the target metals? Or were they chosen de to their location (near motorways, industrial plants, dumps, etc.)? Probably, authors gave detailed information in the reference paper, but it would be good to do this (briefly) in this section too.

Line 110: Please, add description of the isolation of Trichoderma strains from samples or any reference. Do not forget that your description of materials and methods should provide a possibility to reproduce your experiments by any reader.

Line 112: please, give a reference confirming the high E/T value may be used to determine the origin of metal contamination.

Line 113: which criteria did you use to select strains for metal tolerance tests?

Line 116: when you write about pure glycerol, do you mean the long-term storage in a freezer? Please, give some details. In addition, you did not mention any information concerning nutrient media you used for isolation, identification, and maintenance of strains, though this information is essential.

Lines 131-132: it would be good to add information about the maximum allowed values of these metals.

Line 146: how did you collect fungi from Petri plates? Please, give a brief explanation of the sample preparation process: which instruments did you use to grab the mycelium, did you put it into any buffer to prepare a sample for the analysis, etc.

HOW did you perform a statistical treatment of the obtained results? There is no information in this section.

RESULTS

 Table 1: "...maximum concentrations are highlighted in shadow and minimum in bold font" - Actually, I see only bold-font marking in the table. Please, check and correct. I also consider the shadow marking will not be good; probably you can use underlined or italic text instead.

Tables 2 and 3: if you arranged the experiment in three replications, then you should show the results in the form of the "mean +/- standard error" or using the least significant difference. In the current form, both tables do not give statistically treated results.

Lines 182-186: I would recommend to mention strain ID together with the sample numbers, since this information would help a reader to understand, which strains were isolated from samples with the low metal content, and which strains from the soil with a high metal content. without this information, it is hard to understand the meaning of the results listed in Tables 2 and 3.

Lines 208-209: do authors consider that the use of a strain with the Ni tolerance at the level of 60 ppm is really promising in view of the existing remediation value (is it the maximum permissible level?) for Ni at the just slightly lower level (50 ppm)?

Table 4: Actually, this table is not necessary since no significant differences were observed. In this case, it is quite enough to describe this result in the text.

Table 6: (1) the title of the table should be self-explaining, so any who see the table only could understand the sense of the data. Please, change it. (2) in the text below the table authors mention strains grown on metal-containing medium. However, the table does not contain this information, i.e., is not self-explaining again. Please, indicate at which metal concentrations (and which metal)  the strains included into the table were grown.

Lines 344-348: according to the author's conclusions, the studied strains demonstrated both biosorption and accumulation of heavy metals that provides their good bioremediation potential. It would be good to make some more explanations how they can be used. Trichoderma are soil fungi, which live, reproduce, and die in the soil. If they are capable of accumulating or biosorbing of Ni and Cu, then after the death of the cells, their decay will just return these metals into the soil, so their use as the tools for soil remediation should include any special methods intended to the irreversible removal or transformation of contaminants. It would be good to add such information with references (or at least authors' hypotheses) at the end of the discussion.

CONCLUSIONS

 "...XRF method revealed accumulation of both Cu and Ni in the examined Trichoderma strains." - Since authors did not describe this method in the Materials and Methods section, it is difficult to understand, how well was this conclusion substantiated. The target metals could bind with the cell membrane and do not enter the cell. Please, add the description of the sample preparation in the corresponding subsection of the Materials and Methods section to show that the possible sorption of metals on cell membranes outside the cells was excluded, and you analyzed namely the intracellular content of these metals.

Author Response

Dear reviewer,

Thank you very much for your detailed review. We have revised the manuscript according to your comments and requirement of the journal. Revised portions are marked in yellow in the paper.

Reviewer 2 Report

A neat study exploring the ability of strains of the same species to grow in the presence of Ni and Cu and giving some information on the mechanism for metal tolerance /uptake. I have a few questions and points to improve clarity:

88: "Green analytical methods?" This is not a phrase I have heard before and I'm not sure it is relevant here. Yes, FTIR and XRF are quick and relatively low cost (as long as you have access to the necessary equipment) but unless there is some reason why a lower cost is preferable in this instance it reads a little like the author used the cheapest/easiest option rather than the best tools for the job. I would leave this section out (lines 88-94) or re-phase if you want to make the point that these easier options yield just as good data as the more involved methods.  

146 - "removed” from the petri dish rather than 'grabbed' 

164-165: This sentence does not make sense. “The concentration of total Cr was two times higher of remediation value, and the concentration of total Ni was 5 times higher of remediation value” Do you mean “The concentration of total Cr was two times higher than the remediation value, and the concentration of total Ni was 5 times higher than the remediation value?

FTIR section: Differences in the FTIR spectrum appear to me to be small and I have a questions about the control cell group. Which strain is the control spectra taken from? Why does the spectra for strain 22665 look different to the others for Nickel?

It would be more convincing if there was a control per species (e.g. control not grown with copper, and control grown with copper) do you have this data. Why not do a control spectra for each species? Or was this done and found to look the same? In my experience FTIR can be highly dependent on the user (e.g. background subtraction) so the more information the better. Perhaps include as supporting information if wanting to reduce the number of spectra for the paper?

Author Response

(The authors gave the same response as above.)

Reviewer 3 Report

The manuscript is of interest for metal bioremediation using fungi. However, there are several aspects that should be carefully revised. In general, all sections of the manuscript should be improved, clarified or deepened, in particular, the introduction and the discussion. Materials and methods section should be improved, as well. There is no statistical analysis, that should be added to show significant differences of obtained results. English and Style editing is required. Authors should also better highlight the novelty elements of their manuscript.

Detailed comments are provided to authors below.

Abstract

L20: “extractable”: with what? Please, clarify. Are you referring to a metal bioavailable fraction? Please, integrate in the text.

L24: please, explain why you have chosen fungal strains, considering the lowest ratio E/T.

L26: please, explicit first the acronym’s meaning, i.e. Fourier-transform infrared spectroscopy (FTIR).

L27: please, define the acronym DTPA and clarify.

L28: please, explain the meaning of “remediation values”, since it is not clear.

In the abstract, authors could better highlight the importance of Trichoderma species/strains from an ecological and biotechnological point of view; moreover, they could better highlight the choice of the FTIR and XRF, as non-destructive and “green” analytical methods, since they mentioned these aspects later in the manuscript.

Introduction

L34-36: Please, consider that in some cases and for natural reasons, metals and metalloids can occur in soils at very high concentrations, becoming a risk for humans and other organisms.

L36-37: “In this case… heavy metals”: please, revise this sentence, since it is not clear. Are you referring to waters, as well?

L37-43: I suggest improving this part, also considering the effects of potentially toxic elements to soil biodiversity, which provide very important ecosystem services for human wellbeing and ecosystem functionality.

L43: I suggest replacing “microbes” with “microorganisms”.

L43-57: I suggest revising this part, improving the structure and deepening the content. Authors should introduce the importance of fungi in bioremediation, describing the most important features and potentialities with the available reviews on this topic in the literature.

L47-49: ”It is…reaction”. Please, revise this sentence, introducing also other involved mechanisms, mentioned later in this section (see L52-56).

L50-51: please, explain better the reason why the polluted soils are important sources of metal-resistant microorganisms.

L63: I also suggest focusing on fungal cell wall. Please, revise.

L66-69: Please, revise this part, since it is not clear. Moreover, this paragraph on the genus Trichoderma could be deepened, providing the most important features and the ecological roles that Trichoderma fungal species play in soils, and potentialities in bioremediation.

L71: please, check “mg/L”.

L85: please, explain the term “available”, since it is not clear.

L88: “green”: please, clarify better this aspect in the introduction.

L97: authors should summarize and provide the main aims of their work.

Materials and methods

L100-102: I suggest providing a complete description of soil sampling and locations.

L108: please, provide details and references for the method used for ICP-OES.

L111: please, explain why the lowest ratio was considered and provide references for this part.

L113-115: which strains are related to the highest ratio E/T and which with the lowest?

L116-117: please, explain in which medium fungal strains were maintained.

L118-123: please, explain how morphological characterization is related to metal tolerance. The reference [27] is quite old. Have you considered also other references for the morphological characterization of Trichoderma? Which were the tested metal concentrations? Please, provide a more detailed description of the applied method, since it is not clear.

L129: please, use other concentration units, e.g. mg/L or M, instead of ppm. Please, revise this in all manuscript. Moreover, the chemical formula is not correctly formatted.

L132: please, provide the concentration values and a reference.

L145-155: please, provide references for used methods.

There is no statistical analysis of the obtained results to define statistical significance, such as for example ANOVA test. I strongly suggest implementing this part. Please, revise.

Results and discussion

L158: please, delete “spp.”, as it is not necessary.

L165-166: please, revise this part since it is not clear.

L167-168: since soil samples under investigation are from vineyards, this aspect should be highlighted also in the introduction, i.e. when authors refer to LUCAS database.

L170-178: authors could explain the reason why other potentially toxic elements are shown along with the two investigated metals.

L182-186: Please, revise this sentence, since it is not clear.

L205-211: Please, improve this part of discussion, also trying to explain the observed “stimulative effect” and the inhibition concentrations. Please, compare the obtained results with those from the available studies in literature and provide references. Is there a relationship between the choice of strains according to the E/T ratio values and the obtained results on metal tolerance’ Please, revise.

Section 3.1.:  how can this part contribute to the metal tolerance/resistance evaluation of tested fungal strains? Please, clarify and in case improve this part.

 L294-302: please, set fungal species/genera names in italics and correct the chemical formula format. Moreover, please, improve this discussion, deepening other aspects related to obtained results.

L329, 351, 355, 360: see comment above.

References

Please, revise all references, checking the required format. In some cases, names of authors are reported, instead of their surnames. Please, set in italics the species/genera names.

Figures

Please, revise the captions of Figures 2 and 3 and check figures resolution.

Author Response

(The authors gave the same response as above.)

Round 2

Reviewer 3 Report

The authors have improved the manuscript. Some further comments are reported below. Please, consider the revised version of the manuscript with track changes.

Line 71: the sentence is not complete. Please revise.

Line 90: Aspergillus spp.

L122: 300: species? please, specify.

L158: please add “the” before superlatives. 

L196: please add the shaker model and brand.

L281-283: which data have you analyzed? Please specify.

Table 1: Data were not statistically analyzed. Why?

Table 2 and 3: you should use an ANOVA test for statistical analysis for repeated measures, since you have done more measures over a time period (3 times period) with the same samples. Please, verify and revise accordingly.

L344-348: these sentences are not clear. Please, explain better and deepen.

Have you considered the possibility of "eustress" responses for the tested fungi to the metal presence?

The stimulation of growth can be a response to metal stress in order to avoid toxic effects.  Please consider papers on this topic in literature, for example, the papers of prof. Gadd and prof. Fomina on fungal growth responses to the presence of metals (e.g. doi: https://doi.org/10.1017/S095375620300786X).

L538-547: I suggest to better connect this part with the previous one, better discussing the importance of fungi in microbial assisted phytoremediation of soils contaminated with potentially toxic elements.

Author Response

Dear reviewer,

Thank you very much for your further comments to our paper. We are grateful for your dedication and time for the revision of our manuscript. Your comments were on spot and helped to clear points we were not aware of.
